# Graph Contrastive Learning Reimagined: Exploring Universality

## ABSTRACT

Graph Contrastive Learning (GCL) presents a promising training paradigm for addressing the label scarcity problem on real-world graph data. Despite its outstanding performance demonstrated in such classical web network tasks as link prediction, its generality to heterophilous networks such as marriage networks has yet to be thoroughly explored. The major factors constraining its generalizability are the encoders and positive sample collection which follow the strong homophilous assumption, which conflicts with the requirements of heterophilous graphs. The logical thought would be to equip GCL with an encoder with learnable propagation weights or generate a more homophilous graph for the input graph. However, the former is experimentally verified to be infeasible and the latter is prohibitive due to self-supervised learning. Therefore, we reaffirmed that the primary cause for its failure is the blind positive sample collection and the cross-layer decay of pseudo-supervised information. To alleviate the above shortcomings, We investigate the characteristic that homophilous graph structure has: i.e., its matrices satisfy the block-diagonal property. Based on this, a new graph contrastive learning framework with an inference module for block diagonal graph structures is proposed, called gRaph cOntraStive Exploring uNiversality (ROSEN), which constructs such structures by learning the local subspace correlations between nodes and their neighbors. It is then applied to the optimization process of contrast loss to aid in the selection of reliable positive samples from the neighborhood and to the encoder process to guarantee the generation of discriminative node representations, respectively. In order to obtain mutually beneficial information for graph structure inference and contrast loss optimization, these two important processes are updated alternately. Thus, theoretically, ROSEN follows the expectation-maximization algorithm. Extensive evaluations of real-world graphs, especially those with heterophilous, have shown the excellent performance and robustness of ROSEN.

## KEYWORDS

Graph Neural Networks; Contrastive Learning for Web Graphs; Graph Representation Learning

### ACM Reference Format:
Anonymous Author(s). 2018. Graph Contrastive Learning Reimagined: Exploring Universality. In *Proceedings of Make sure to enter the correct conference title from your rights confirmation emai (Conference acronym 'XX).* ACM, New York, NY, USA, 11 pages. https://doi.org/XXXXXXX.XXXXXXX

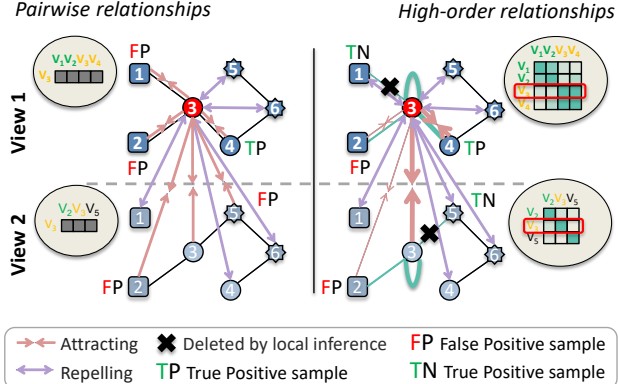

**Figure 1: Comparison of localized GCL and the proposed localized GCL. The thickness of the line indicates the weight.**

## 1 INTRODUCTION

Self-supervised Graph Learning (SSGL), which provides practical guidance for network training by mining implicit pseudo-supervised information, has emerged as a standard unsupervised paradigm for representing Web graph [22, 38]. As one of the most concerned SSGL, Graph Contrastive Learning (GCL) proposes a more essential and intuitive design that captures the invariance information via consistency maximization of two graphs (views) and has been utilized in diverse Web applications, such as recommender systems [15, 20]. When it comes to the source of self-supervised information, most GCL methods follow a heuristic approach presented for contrastive learning in computer vision [5, 8], in which the same nodes in the augmented graphs are set as positive samples of each other [44], named **Paired GCL**. In addition, GCL provides a graph-specific scheme based on the homophilous assumption, namely appending the neighbor nodes into the positive samples, called **Localized GCL**, as shown in Figure 1 (a). These schemes contribute excellent expert knowledge for mining pseudo-supervised information, resulting in efficient and stable GCL frameworks [13, 19].

Unfortunately, most existing GCL frameworks still fail to cover the requirements for universality. To be specific, real-world web graphs are diverse, instead of just those linking the same-type website, such as heterophilous networks [39], which undoubtedly put forward a request for a universal GCL framework. However, the above GCL frameworks are powerless for this request since their key components (i.e., GNN encoder [18, 37] and contrastive loss [19]) follow the strong homophilous assumption, which is considered a major factor of failure on the heterophilous graphs [6, 43]. Thus, a straightforward solution for this drawback may be to equip an elaborated GNN encoder that has verified the effectiveness of these complex web graphs in supervised scenarios, such as GAT [32] and FAGCN [2]. Yet such a strategy has been experimentally

verified to be suboptimal on heterophilous web graphs. We believe its main causes are: blind positive sample collection and the decay of the pseudo-supervised information.

On the one hand, the propagation induced by contrastive loss optimization is equivalent to jointly feature smoothing [4] (i.e., enlarge similarity) and sharpening [40] (i.e., decline similarity) on the positive and negative pairs, respectively [35]. Based on the requirement for label consistency [36], the ideal positive sample sets for the localized GCL scheme should be composed of only neighbor nodes whose class are same as the center nodes (TRUE positive samples) instead of all neighbor nodes. Training with positive samples that are collected blindly, the network parameters of Localized GCL would not fit the true data distribution accurately, resulting in a node representation with unauthentic predictions. As key pseudo-supervised information, the indicator matrix that points to TRUE positive samples supplies the correct direction for feature updating. However, since the labels are unknown, it is a tough challenge to construct such an indicator matrix.

On the other hand, the decay of cross-layer pseudo-supervised information is also responsible for the mistakes in propagation weight training. Specifically, the propagation within the encoder takes its orientation (i.e., smoothing or sharpening) with the help of learnable propagation weights. As pseudo-supervised information for training propagation weights, positive sample pairs identify whether the nodes at both ends of the edges are similar. Typically it directly affects the learning of the outermost propagation weights while ignoring the correctness of the learning of the inner propagation weights, which tends to decay the pseudo-supervised information, thereby disrupting parameter training. Actually, for achieving label-guided classified propagation, several Graph Neural Network (GNN) models have been provided, such as CPGNN [42] and BM-GCN [12]. The former propagates soft labels under the guidance of a compatibility matrix, while the latter propagates features over a block matrix built with soft labels. Still, the problem remains in unsupervised settings.

To remedy the two drawbacks, this study attempts to devise a self-supervised structure inference module to adapt the GCL to diverse web graphs. Firstly, a theoretical analysis reveals the graph structure matrices corresponding to the fully homophilous graph must meet the Block Diagonal Property. On this basis, the structure inference module can pursue the homophilous graph structure by implementing the Block Diagonal Representation algorithm on the feature space. The inferred graph structure matrix approximately characterizes the higher-order relationships between same-subspace nodes and may be an informative indicator for collecting TRUE positive samples and learning correct propagation weights in the GNN encoder. Yet, this characterization might not be precise owing to the presence of massive data noise.

To meet this challenge, this study proposes a novel graph contrastive learning framework, named gRaph cOntraStive Exploring uNiversality (ROSEN), which leverages two strategies, Local Feature Space Training, and Alternating Update, to improve the effectiveness of the structure inference module progressively. Specifically, synchronously calculating the block diagonal matrices on the Ego network feature space of the nodes, and then extracting the vectors of the corresponding nodes to reconstruct the graph

structure, as shown in Figure 2.(b). Aiming at the blindness problem of positive sample collection for local graph contrast learning, ROSEN assists contrast learning in judging reliable positive samples (i.e., the neighbor nodes in the same subspace as the center node) based on the correlation of local subspaces. In addition, the inferred graph structure is directly incorporated into the feature propagation at each layer of the encoder as a solution for the problem of cross-layer pseudo-supervised information decay. To provide low-noise features for graph structure inference and credible graph structures for contrastive loss optimization, these crucial components ROSEN are optimized alternatively to obtain reciprocal information. Theoretically, it is proven that ROSEN can be formulated as the maximum likelihood of nodes and neighbor nodes, which is resolved by the Expectation Maximization (EM) algorithm. The main contributions of this study are summarized as follows

- We investigate the block diagonal properties presented by the homophilous graph structure matrix.
- We propose a generalized graph contrast learning framework with graph structure inference, named gRaph cOntraStive Exploring uNiversality (ROSEN).
- We theoretically prove that the proposed local graph contrast learning framework follows the EM algorithm.
- We perform intensive experiments on publicly available datasets to provide evidence for the effectiveness and universality of the ROSEN.

## 2 PRELIMINARIES

In this section, we commence by introducing notations used throughout the paper. And then we elucidate the basic concepts of Graph Contrastive Learning (GCL).

### 2.1 Graph Contrastive Learning

**Notations.** Let $\mathcal{G}(\mathcal{V}, \mathcal{E}, \mathbf{X})$ denotes an attributed graph, where $\mathcal{V}$ is the node set, $\mathcal{E}$ is the edge set, and $\mathbf{X} \in R^{N \times F}$ represents the node attribute matrix, where $N$ and $F$ are the number of nodes and attributes, respectively. The adjacency matrix is denoted by $\mathbf{A} \in R^{N \times N}$. The laplacian matrix is defined as $L_\mathbf{A} = Diag(\mathbf{A1}) - \mathbf{A} = \mathbf{D} - \mathbf{A}$, where $\mathbf{D}$ denots the diagonal degree matrix. Based on the edge set $\mathcal{V}$, the neighbor node set of each node can be obtained (e.g., $N(v)$ of node $v$). The label matrix is denoted by $\mathbf{Y} \in R^{N \times C}$, which only is employed in fine-tune the parameters of the classifier on downstream tasks, such as node classification.

**GNN Encoder.** To produce informative node representations, the raw attribute is projected by a shared encoder with learnable propagation weights (e.g., Graph Attention Network (GAT) [32]). For the given graph $\mathcal{G}(\mathcal{V}, \mathcal{E}, \mathbf{X})$, this encoding process can be formulated as

$$GCN(\tilde{\mathbf{A}}, \mathbf{H}^{(l)}) : \mathbf{H}^{(l+1)} = \sigma(\tilde{\mathbf{A}} \mathbf{H}^{(l)} \mathbf{W}^{(l)}) \quad (1)$$

where $\mathbf{H}^{(0)} = \mathbf{X}$ is the initial representation, and $\sigma$ terms the non-linear activation function (e.g., ReLU), and $\mathbf{W}^{(l)}$ represents the parameter matrix. Consequently, by encoding node via GNNs, node representations would capture the structural relationships and local patterns on graphs.

**Paired and Localized GCL.** Based on the defined positive and

negative sample sets, contrastive loss is implemented as the distance minimization between the positive pairs alongside the distance maximization between the negative pairs. As exemplified by a node-level GCL scheme, a widely used InfoNCE [30] loss $\mathcal{L}_{info}$ can be described as follows:

$$\mathcal{L}_{info} = -\frac{1}{|V|} \sum_{v \in V} log \frac{pos(v)}{pos(v) + neg(v)}$$

$$pos(v) = \sum_{v^+ \in \mathcal{P}_v} e^{\theta(\mathbf{h}_v, \mathbf{h}_{v^+})/\tau}, \quad neg(v) = \sum_{v^- \in \mathcal{N}_v} e^{\theta(\mathbf{h}_v, \mathbf{h}_{v^-})/\tau} \quad (2)$$

where $\theta$ represents the cosine similarity. $\tau$ is a temperature coefficient, and smaller one help form a more uniform representation space. $\mathcal{P}_v$ and $\mathcal{N}_v$ denote the positive and negative sample set of node $v$, respectively. The positive sample selection strategies utilized in GCL include two categories: Paired positive smaples and Localized positive samples. Generally, the former utilizes the same node in another graph [44], while the latter append nodes with similar semantics [13, 19]. Based on the homophilous assumption [1, 25], i.e., connected nodes have similar semantics, neighbors on the graph structure are considered as credible positive samples. As a consequence, they can be expressed as

$$Paired \quad \mathcal{P}_v = \{u|u = \tilde{v}\}, \quad (3)$$

$$Localized \quad \mathcal{P}_v = \{u|u \in N(v) \text{ and } u \in \{N(\tilde{v}) \cup \tilde{v}\}\} \quad (4)$$

Negative sample set contains all the remaining nodes, i.e., $\mathcal{N}_v = \{\{V \cup \tilde{V}\} \backslash \{\mathcal{P}_v \cup v\}\}$. Compared to the paired GCL, the localized one enhances the characterization of homophily, which is regarded as a reliable guide of the feature propagation in unsupervised scenarios [19], resulting in a accurate representation of inductive biases on real-world Web graphs [26].

## 3 METHODOLOGY

Responding to the dilemma of Graph Contrastive Learning (GCL) mentioned in the introduction, this section begins by proving the efficacy of block diagonal graph structure matrices for this problem theoretically. Then, a novel framework with block diagonal structure inference for graph contrast learning is proposed, named gRaph cOntraStive Exploring uNiversality (ROSEN), and benefits from the alternating update of graph structure inference and contrast loss optimization. Lastly, a theoretical explanation of the proposed framework ROSEN is provided, from an expectation-maximization perspective.

**Motivation.** An efficient solution to the problems presented in the Introduction is constructing a fully homophilous graph structure by removing the edges linking the node pairs of different classes in the initial graph structure. However, getting such a graph structure by graph augmentation [28, 41] is tricky since the node labels are unknown in self-supervised learning scenarios. To overcome this challenge, this study first investigates the properties the matrix of the entire homophilous graph structures possessed.

*Definition 3.1.* Block Diagonal Property [14]. Given a square matrix, if it can be partitioned into multiple small block matrices, where each block is represented by non-zero elements on the principal diagonal and all non-diagonal elements are zero, it obeys the block diagonal property.

THEOREM 3.2. *For the permutation matrix $\mathcal{P}$, which can be utilized in an elementary row transformation to make the attribute matrix $\mathbf{X}$ to be sorted according to class, after elementary row-column transformation using $\mathcal{P}$, the adjacent matrix $\mathbf{S}$ of fully homophilous graph matrices will definitely satisfy the Block Diagonal Property explained in Define 3.1.*

PROOF. Firstly, the permutation matrix $\mathcal{P}$ can be viewed as a node sorting operator since the transformed matrix after class sorting can always be denoted as

$$\hat{\mathbf{X}} = \mathcal{P}\mathbf{X} = [\hat{\mathbf{X}}_1, \hat{\mathbf{X}}_2, \ldots, \hat{\mathbf{X}}_K],$$

where $\hat{\mathbf{X}}_i \in R^{n_i \times F}$ denotes the feature sub-matrix of the nodes for class $i$, and having $\sum_{i=0}^{C} n_i = N$. Therefore, the transformed matrix $\hat{\mathbf{S}} = \mathcal{P}\mathbf{S}\mathcal{P}^\top$ is a matrix sorted according to class. Since there are only the edges linking the same class of node pairs in the fully homophilous graph, the transformed adjacent matrix $\hat{\mathbf{S}} = \mathcal{P}\mathbf{S}\mathcal{P}^\top$ can always be formulated as

$$\begin{bmatrix} \hat{\mathcal{S}}_1 & 0 & \cdots & 0 \\ 0 & \hat{\mathcal{S}}_2 & \cdots & 0 \\ \vdots & \vdots & \ddots & \vdots \\ 0 & 0 & \cdots & \hat{\mathcal{S}}_K \end{bmatrix} \quad (5)$$

where $\hat{\mathcal{S}}_i \in R^{n_i \times n_i}$ is a block matrix, which describes the relations between the nodes of class $i$. Thus, it is can be concluded that the adjacent matrix $\hat{\mathcal{S}}$ satisfy the Block Diagonal Property. □

### 3.1 ROSEN Framework

Based on the analysis presented in the previous section, this study intends to propose a structure inference module for WGCL in pursuit of Block Diagonal graph structures. Inspired by the classic Block Diagonal Representation (BDR) [23], the module can generate coefficient matrix that approximate block diagonal by optimizating subspace clustering with soft diagonal block regularization on feature space, namely

$$\min_{\mathbf{B}} \frac{1}{2} \|\mathbf{X} - \mathbf{BX}\|^2 + \gamma \|\mathbf{B}\|_{\underline{k}}$$

$$\text{s.t. } \text{diag}(\mathbf{B}) = 0, \mathbf{B} \geq 0, \mathbf{B} = \mathbf{B}^T \quad (6)$$

where $\mathbf{B}$ stands for the coefficient matrix, and $\|\mathbf{B}\|_{\underline{k}} = \sum_{i=0}^{k-1} \lambda_i (\mathbf{L_B})$ represents the sum of the smallest $i$ eigenvalue of the laplacian matrix of $\mathbf{B}$, where the eigenvalues are listed in ascending order, and $\gamma$ is a scalar for tradeoff between two terms. These two terms of Equation (7) guarantee that matrix $\mathbf{B}$ is self-expressive [21] and has $k$ connected subgraphs (i.e., blocks) [7].

In the proposed module, three constraints for matrix $\mathbf{B}$ include no self-loops, non-negativity, and symmetry. However, applying these constraints directly to $\mathbf{B}$ would limit its expressive power. Thus, this module seeks to introduce an approximation term to alleviate these constraints, namely rephrasing Equation (6) as

$$\min_{\mathbf{Z}, \mathbf{B}} \frac{1}{2} \|\mathbf{X} - \mathbf{ZX}\|^2 + \frac{\lambda}{2} \|\mathbf{Z} - \mathbf{B}\|^2 + \gamma \|\mathbf{B}\|_{\underline{k}},$$

$$\text{s.t. } \text{diag}(\mathbf{B}) = 0, \mathbf{B} \geq 0, \mathbf{B} = \mathbf{B}^T \quad (7)$$

where $\mathbf{Z}$ denotes the coefficient matrix for approximating $\mathbf{B}$. This objective function can be optimizaed via alternating minimization

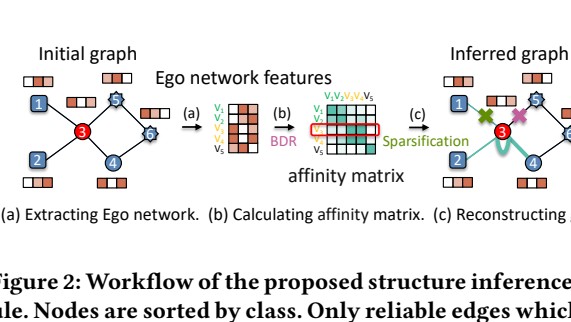

(a) Extracting Ego network.  (b) Calculating affinity matrix.  (c) Reconstructing graph

**Figure 2: Workflow of the proposed structure inference module. Nodes are sorted by class. Only reliable edges which connect the node pairs belonging to the same subspace are kept.**

solver [34], namely, update one while fixing the other. After the solver converges, the generated coefficient matrices would be applied in the ROSEN training. Even though it appears plausible to generate reliable graph structures for ROSEN leveraging this proposed module, the process would be hampered by the massive **data noise** in real-world graphs, resulting in the generated structures being off the Block Diagonal Property.

To tackle this drawback, this study proposes a novel contrastive learning framework for web graphs, named gRaph cOntraStive Exploring uNiversality (ROSEN). It leverages two strategies, Local Feature Space Training and Alternating Optimization, to improve the effectiveness of the graph structure inference module progressively, thereby generating discriminative node representations.

*3.1.1 Local Block Diagonal Graph Structure Inference.* Since nodes in the neighborhood determined by linking have more substantial relevance than those out of the neighborhood, the structure inference module is set up on the node local subgraphs to minimize the impact of data noise. In line with extract-then-calculate, "Local Feature Space Training" strategy focuses on synchronously optimizing the following objective function on the obtained node Ego networks (i.e., combinations of nodes and their one-hop neighbors) to compute affinity matrices.

$$\min_{\mathbf{Z}_v, \mathbf{B}_v} \frac{1}{2} \|\mathbf{H}_v - \mathbf{Z}_v \mathbf{H}_v\|^2 + \frac{\lambda}{2} \|\mathbf{Z}_v - \mathbf{B}_v\|^2 + \gamma \|\mathbf{B}_v\|_{\underline{k}},$$
$$s.t. \ \mathrm{diag}(\mathbf{B}_v) = 0, \ \mathbf{B}_v \geq 0, \ \mathbf{B}_v = \mathbf{B}_v^T \tag{8}$$

where $\mathbf{H}_v \in R^{(|N(v)|+1)\times D}$ denotes the feature matrix for the Ego network of node $v$. Additionally, $\mathbf{Z}_v$ and $\mathbf{B}_v$ represents the coefficient matrices for this Ego network, which are abbreviated as $\mathbf{Z}$ and $\mathbf{B}$ in the calculations for ease of presentation. Since $\|\mathbf{B}\|_{\underline{k}}$ is nonconvex term for which the optimization is NP-hard, we convert it into a convex program using the properties about the sum of eigenvalues [3]

$$\|\mathbf{B}\|_{\underline{k}} = \min_{\mathbf{W}} \langle \mathbf{L_B}, \mathbf{W} \rangle, \text{ s.t. } 0 \leq \mathbf{W} \leq \mathbf{I}, \mathrm{Tr}(\mathbf{W}) = k \tag{9'}$$

where $\mathbf{W}$ is the newly added variable block. Therefore, the overall objective for the proposed structure inference module with these three variable blocks (i.e., $\mathbf{Z}_v$, $\mathbf{B}_v$, and $\mathbf{W}_v$) is

$$\min_{\mathbf{Z},\mathbf{B},\mathbf{W}} \frac{1}{2} \|\mathbf{H}_v - \mathbf{Z}\mathbf{H}_v\|^2 + \frac{\lambda}{2} \|\mathbf{Z} - \mathbf{B}\|^2 + \gamma \langle \mathrm{Diag}(\mathbf{B1}) - \mathbf{B}, \mathbf{W} \rangle$$
$$\text{s.t. } \mathrm{diag}(\mathbf{B}) = 0, \mathbf{B} \geq 0, \mathbf{B} = \mathbf{B}^\top, 0 \leq \mathbf{W} \leq \mathbf{I}, \mathrm{Tr}(\mathbf{W}) = k \tag{9}$$

By utilizing alternating minimization, this objective function is split into three problems and solved independently, as described in

the appendix. Ultimately, the closed solutions for the three blocks are derived as

$$\mathbf{W}^{k+1} = \mathbf{U}\mathbf{U}^\top, \tag{10}$$

$$\mathbf{Z}^{k+1} = \left(\mathbf{H}_v^\top \mathbf{H}_v + \lambda \mathbf{I}\right)^{-1} \left(\mathbf{H}_v^\top \mathbf{H}_v + \lambda \mathbf{B}\right) \tag{11}$$

$$\mathbf{B}^* = \left[\left(\hat{\mathbf{A}} + \hat{\mathbf{A}}^\top\right)/2\right]_+ \tag{12}$$

where $\mathbf{U} \in \mathbb{R}^{(|N(v)|+1)\times k}$ contains the eigenvectors matching the $k$ smallest eigenvalues of the laplacian matrix $\mathbf{L_B}$. Additionally, $\hat{\mathbf{A}} = \mathbf{A} - \mathrm{Diag}(\mathrm{diag}(\mathbf{A}))$, where $\mathbf{A} = \mathbf{Z} + \frac{\gamma}{\lambda}\left(\mathrm{diag}(\mathbf{W})\mathbf{1}^\top - \mathbf{W}\right)$. Please consult Appendix for the alternating minimization solver of Equation (9).

Following two coefficient matrices for the Ego network are obtained, the row vectors and indices corresponding to the center nodes are extracted from them to build two affinity matrix (denoted by $\mathbf{A_Z}$ and $\mathbf{A_B}$) describing the full graph structure. Moreover, to guarantee matrix non-negativity and symmetry, we construct the adjacency matrix $\mathbf{S}$ by taking absolute values and applying symmetry operations. This can be formulated as

$$\mathbf{S} = \left(|\mathbf{A_B}| + |\mathbf{A_B}^\top|\right)/2 \quad \text{or} \quad \mathbf{S} = \left(|\mathbf{A_Z}| + |\mathbf{A_Z}^\top|\right)/2 \tag{13}$$

Moreover, to ease the computational burden and preserve confident relationship, a sparse graph structure is generated by a matrix sparsification operation, i.e., zeroing of matrix values less than a threshold $\beta$, which can be formulated as

$$\mathbf{S}_{ij} = \begin{cases} 0, & \text{if } \mathbf{S}_{i,j} < \beta, \\ \mathbf{S}_{i,j}, & \text{otherwise} \end{cases}$$

**Benefits.** The proposed structure inference module for web graph contrastive learning provides the below advantages. (1) Robust topology augmentation. As only non-zero valued edges are meaningful, those tend to link pairs of nodes belonging to the same subspace will be preserved by this module, i.e., robust edge deletion, which would offer reliable positive samples as pseudo-supervised information for WGCL. (2) Higher-order information. Given that the inferred edge weight favors the characterization of higher-order relationships between local neighbors, it stands for the inferred graph structure will boost the representation capability of the WGCL framework by exploring mesocosmic community structure rather than fragile pairwise relationships. Furthermore, the module performs parallel training on nodes, which may facilitate the scalability of the framework.

*3.1.2 Contrastive Learning Optimization.* After a more credible graph structure matrix is constructed, the discussion follows on how to improve the generality of graph contrastive learning. Firstly, the proposed ROSEN proposes leveraging this matrix to guide the optimization process of localized graph contrastive learning. To be specific, given node features $\mathbf{H}$ and inferred graph structure matrix $\mathbf{S}$, it devises the objective function $\mathcal{L}_{bdgcl}$ for node $v$ as

$$-log \frac{\sum\limits_{v^+ \in N_v^S} \mathbf{S}_{v,v^+} * e^{\theta(\mathbf{h}_v, \mathbf{h}_{v^+})/\tau}}{\sum\limits_{v^+ \in N_v^S} \mathbf{S}_{v,v^+} * e^{\theta(\mathbf{h}_v, \mathbf{h}_{v^+})/\tau} + \sum\limits_{v^- \in \{V \setminus \mathcal{N}_v^S\}} e^{\theta(\mathbf{h}_v, \mathbf{h}_{v^-})/\tau}} \tag{14}$$

where $N_v^S$ denotes the neighbor node set of node $v$, which is determined by the matrix $\mathbf{S}$. In contrast to existing localized contrast

scheme which blindly assuming all neighbor nodes are the positive samples [13, 19], the proposed ROSEN elaborately selectes the reliable neighbor nodes as TRUE positive samples through the local structure inference module. In addition, the local higher-order relationships provided by weights could efficiently supplement the fragile pairwise relationships between the sample pairs in contrast learning computations.

**Alternating update.** Considering that graph neural networks (GNNs) can essentially serve as denoising encoders [24], which meets the demand of low-noise features for the proposed structure inference module, the construction of graph structure matrices on the encoded features is a logical solution. To provide expressive encoded features, the proposed ROSEN presents the alternating optimization scheme of the encoder parameters and graph structure. Additionally, the inferred graph structure is adopted for the encoding process to alleviate the discriminative information loss of the node features [12, 42].

THEOREM 3.3. *Let $\Theta$, $k$ and $\mathbb{1}G$ denote the parameters of the GNN encoder, the number of subspaces (blocks), and the subspace indicator, respectively, the proposed ROSEN follows the Expectation-Maximization (EM) algorithm, in which the structure inference and the maximization of the lower bound on the mutual information of the contrastive pairs' representation are equivalent to the E step and the M step, respectively.*

Please refer to the appendix for the detailed proof.

## 4 EXPERIMENTS

In this section, we begin by introducing the fundamental setups of the experiment, including datasets, comparison methods, and configurations. Then, to comprehensively assess the proposed ROSEN, the performances on two mainstream tasks, i.e., node classification and clustering, are experimentally validated. Lastly, several experiments are utilized to provide an intuitive understanding of its performance improvement, including the effectiveness test, the ablation study, the hyperparameter analysis, and the robustness analysis.

## 4.1 Experimental Setup

*4.1.1 Datasets.* To rigorously evaluate the proposed ROSEN, twelve publicly available graph datasets are adopted in the experiment. According to whether the Edge Homophily [26] is more significant than 0.5, these datasets can be divided into two categories: homophilous graph (i.e., Cora, CiteSeer, PubMed, Wiki-CS, Computers, and Photo) and heterophilous graph (i.e., Cornell, Texas, Wisconsin, Chameleon, Squirrel, and Actor). The detailed statistics are in Table 5 in the Appendix.

For a fair model comparison, the datasets are split according to the broadly adopted scheme [26]. Specifically, As for three homophilous graphs (Cora, CiteSeer, and Pubmed) and all six heterophilous graphs, a publicly available split is adopted, which splits 48%, 32%, and 20% of the data are used for training, validation, and testing, respectively. In addition, for three large homophilous graphs (Wiki-CS, Computers, and Photo), we randomly split the training, validation, and test sets into 10%, 10%, and 80%.

*4.1.2 Comparison Methods.* The comparison methods are the following four types, i.e., classical semi-supervised graph neural network (including GCN [18], GAT [32], and JKNetJKNet), and an unsupervised method ( K-Means [10]) and unsupervised graph learning model (DeepWalk [27], Node2Vec [9], GAE [17], VGAE [17]), and the latest graph contrastive learning framework (including DGI [33], MVGRL [11], GRACE [44], GCA [45], BGRL [29], and HomoGCL [19]).

*4.1.3 Configurations.* All experiments are performed on a single GeForce RTX4090 24GB GPU. The results are reported as an average of over ten random runs. The experiment works with the configurations reported in the original paper for all comparison methods. It is worth mentioning that, thanks to the open PyTorch libraries: PyG[1] and PyGCL[2], the reproduction of all comparison methods is facilitated. For the proposed framework ROSEN, the encoder picks a two-layer GCN [18] network, and the dimensions of its hidden layer are selected as 64. The graph augmentation strategies are masking attribute and edge deletion, which have the ratio {0.2, 0.4, 0.6, 0.8}. The temperature coefficient of the contrastive loss is taken from {0.1, 0.3, 0.5, 0.7, 1, 2}. In addition, the network optimizer used during network training is Adam optimizer [16], the learning rate is taken out of {0.001, 0.01} and the weight decay rate is chosen in $\{0, 1\times10^{-3}, 1\times10^{-4}, 1\times10^{-5}\}$. The hyperparameter $\lambda$, $\gamma$, and *tol* used for graph structure learning are chosen among $\{40, 50, 60\}, \{0.5, 1, 2\}$, and $\{1 \times 10^{-3}, 1 \times 10^{-4}\}$, respectively. The number of blocks for subspace clustering is picked from a range no more significant than the number of classes, and its impact on the results is thoroughly analyzed in Section 4.5.

## 4.2 Results and Analysis

**Homophilous datasets.** Table 1 presents the node classification performance of all methods on the homophilous graph. Observing the data presented in Table 1, it is evident that the proposed ROSEN exhibits superior classification performance on all datasets compared to all unsupervised baseline methods. For example, it outperforms the second-ranked MVGRL on the Cora dataset by 0.95%, which demonstrates the effectiveness of the proposed graph structure inference module for graph contrastive learning. Furthermore, even in comparison to the supervised baseline methods, the ROSEN obtains the highest classification accuracy on all the other five datasets except CiteSeer and PubMed. Of note is the performance on the large dataset. It outperformed the runner-up GAT on the Computers dataset by 1.97% and the runner-up JKNet on the Photo dataset by 1.22%. This illustrates that the inferred graph structure provides more precise pseudo-supervised information, which proves the potent data-mining capabilities of the ROSEN.

**Heterogeneous datasets.** As discussed before, the design of most existing unsupervised graph neural networks (especially graph contrastive learning methods) is based on the strong homophilous assumption (i.e., neighbor nodes are treated to be TRUE positive samples), which renders them powerless in tackling heterophilous problem. The performance comparison of the ROSEN and the comparison methods on the heterophilous graph is exhibited in Table 2. It can be observed that, as compared to all unsupervised baseline

---

[1]https://www.pyg.org/
[2]https://github.com/PyGCL

**Table 1: The accuracy in percentage (mean ± std) of node classification for six homophilous datasets. The Best and Runner-up are highlighted by bolding and underlining, respectively. The second column presents the data considered during training.**

| Model | Training Data | Cora | CiteSeer | PubMed | Wiki-CS | Computers | Photo |
|---|---|---|---|---|---|---|---|
| GCN | A, X, Y | 85.77 ± 0.25 | 73.68 ± 0.31 | **88.13 ± 0.28** | 76.89 ± 0.37 | 86.34 ± 0.48 | 92.35 ± 0.25 |
| GAT | A, X, Y | 86.37 ± 0.30 | 74.32 ± 0.27 | 87.62 ± 0.26 | 77.42 ± 0.19 | 87.06 ± 0.35 | 92.64 ± 0.42 |
| JKNet | A, X, Y | 85.93 ± 1.35 | **74.37 ± 1.53** | 87.68 ± 0.30 | 79.52 ± 0.21 | 85.28 ± 0.72 | 92.68 ± 0.13 |
| DeepWalk | A | 73.96 ± 0.12 | 61.91 ± 0.42 | 74.79 ± 0.98 | 74.35 ± 0.06 | 85.68 ± 0.06 | 89.44 ± 0.11 |
| Node2Vec | A | 75.87 ± 0.22 | 62.54 ± 0.13 | 76.49 ± 0.32 | 71.79 ± 0.05 | 84.39 ± 0.08 | 89.67 ± 0.12 |
| GAE | A, X | 76.83 ± 1.22 | 65.43 ± 1.13 | 76.52 ± 0.33 | 70.15 ± 0.01 | 85.27 ± 0.19 | 91.62 ± 0.13 |
| VGAE | A, X | 79.36 ± 0.83 | 69.18 ± 0.27 | 79.17 ± 0.44 | 76.63 ± 0.19 | 86.37 ± 0.21 | 92.20 ± 0.11 |
| DGI | A, X | 85.90 ± 0.57 | 72.57 ± 0.23 | 83.52 ± 1.24 | 75.73 ± 0.13 | 84.09 ± 0.39 | 91.49 ± 0.25 |
| MVGRL | A, X | 86.77 ± 0.33 | 73.71 ± 0.48 | 84.63 ± 0.73 | 77.97 ± 0.18 | 87.09 ± 0.27 | 92.01 ± 0.13 |
| GRACE | A, X | 84.79 ± 0.64 | 72.94 ± 0.72 | 84.51 ± 0.68 | 79.16 ± 0.36 | 87.21 ± 0.44 | 92.65 ± 0.32 |
| GCA | A, X | 85.16 ± 0.51 | 72.73 ± 0.45 | 85.22 ± 0.73 | 79.35 ± 0.12 | 87.84 ± 0.27 | 92.78 ± 0.17 |
| BGRL | A, X | 85.37 ± 0.74 | 73.45 ± 0.83 | 84.61 ± 0.32 | 78.74 ± 0.22 | 88.92 ± 0.33 | 93.24 ± 0.29 |
| HomoGCL | A, X | 85.02 ± 0.68 | 73.67 ± 0.78 | 82.33 ± 0.49 | 77.47 ± 0.45 | 87.84 ± 0.28 | 93.59 ± 0.27 |
| ROSEN | A, X | **87.72 ± 1.00** | 74.13 ± 0.68 | 85.30 ± 0.72 | **80.17 ± 1.28** | **89.03 ± 0.41** | **93.90 ± 1.10** |

**Table 2: The accuracy in percentage (mean ± std) of node classification for six heterophilous datasets. The Best and Runner-up are highlighted by bolding and underlining, respectively. The second column presents the data considered during training.**

| Model | Training Data | Cornell | Texas | Wisconsin | Chameleon | Squirrel | Actor |
|---|---|---|---|---|---|---|---|
| GCN | A, X, Y | 55.14 ± 7.57 | 55.68 ± 9.61 | 58.42 ± 5.10 | 59.82 ± 2.58 | 36.89 ± 1.34 | 30.64 ± 1.49 |
| GAT | A, X, Y | 58.92 ± 3.32 | 58.38 ± 4.45 | 55.29 ± 8.71 | 60.26 ± 2.50 | 40.72 ± 1.55 | 27.44 ± 0.89 |
| JKNet | A, X, Y | 56.49 ± 3.22 | 65.35 ± 4.86 | 51.37 ± 3.21 | **60.31 ± 2.76** | **44.24 ± 2.11** | **36.47 ± 0.51** |
| DeepWalk | A | 39.18 ± 5.57 | 46.49 ± 6.49 | 33.53 ± 4.92 | 47.74 ± 2.05 | 32.93 ± 1.58 | 22.78 ± 0.64 |
| Node2Vec | A | 42.94 ± 7.46 | 41.92 ± 7.76 | 37.45 ± 7.09 | 41.93 ± 3.29 | 22.84 ± 0.72 | 28.28 ± 1.27 |
| GAE | A, X | 58.85 ± 3.21 | 58.64 ± 4.53 | 52.55 ± 3.80 | 33.84 ± 2.77 | 28.03 ± 1.61 | 28.03 ± 1.18 |
| VGAE | A, X | 59.19 ± 4.09 | 59.20 ± 4.26 | 56.67 ± 5.51 | 35.22 ± 2.71 | 29.48 ± 1.48 | 26.99 ± 1.56 |
| DGI | A, X | 63.35 ± 4.61 | 60.59 ± 7.56 | 55.41 ± 5.96 | 39.95 ± 1.75 | 31.80 ± 0.77 | 29.82 ± 0.69 |
| MVGRL | A, X | 64.30 ± 5.43 | 62.38 ± 5.61 | 62.37 ± 4.32 | 51.07 ± 2.68 | 35.47 ± 1.29 | 30.02 ± 0.70 |
| GRACE | A, X | 54.86 ± 6.95 | 57.57 ± 5.68 | 50.00 ± 5.83 | 48.05 ± 1.81 | 31.33 ± 1.22 | 29.01 ± 0.78 |
| GCA | A, X | 55.41 ± 4.56 | 59.46 ± 6.16 | 50.78 ± 4.06 | 49.80 ± 1.81 | 35.50 ± 0.91 | 29.65 ± 1.47 |
| BGRL | A, X | 57.30 ± 5.51 | 59.19 ± 5.85 | 52.35 ± 4.12 | 47.46 ± 2.74 | 32.64 ± 0.78 | 29.86 ± 0.75 |
| HomoGCL | A, X | 48.64 ± 2.59 | 54.05 ± 2.32 | 39.21 ± 5.75 | 48.68 ± 1.16 | 38.71 ± 0.85 | 28.81 ± 0.78 |
| ROSEN | A, X | **76.49 ± 6.84** | **74.86 ± 6.29** | **78.63 ± 4.68** | 49.25 ± 2.33 | 39.13 ± 1.36 | 33.19 ± 0.81 |

**Table 3: Overall performance of node clustering measured by ACC, NMI, and ARI scores in percentage. The best results are in bold, and the second-best results are underlined.**

| | Cora | | | Citeseer | | |
|---|---|---|---|---|---|---|
| | ACC | NMI | ARI | ACC | NMI | ARI |
| K-Means | 35.78 | 16.88 | 8.30 | 44.47 | 21.35 | 17.43 |
| GRACE | 64.02 | 36.17 | 22.69 | 53.65 | 27.62 | 25.14 |
| BGRL | 61.84 | 40.39 | 24.29 | 52.52 | 15.4 | 14.17 |
| MVGRL | 72.45 | 55.05 | 41.55 | 64.14 | 39.12 | 38.93 |
| ROSEN | **76.08** | **58.53** | **46.50** | **66.22** | **40.34** | **40.17** |

methods, ROSEN consistently achieves significant performance improvement on five heterophilous datasets except for Chameleon, which demonstrates the generality and robustness of the designed ROSEN. In particular, on the Cornell, Texas, and Wisconsin datasets,

it is above the classification accuracies of the second-place MV-GRL by 12.19%, 12.48%, and 16.26%, respectively. Besides, it considerably outperforms the supervised baseline methods on these three datasets and achieves comparable performance on the other three datasets. This is mainly attributed to the availability of the homophilous graph structure in the optimization of comparative learning, which confirms the effectiveness of the graph structure inference module.

**Visualization.** To visually compare the representation capability of the GCL framework, the generated node representations are dimensionally reduced and visualized via t-SNE [31]. Figure 3 presents the results for GRACE, BGRL, and the proposed ROSEN over the datasets Cora, Photo, and Wiki-CS. It is concluded from the result that the quality of the node embedding produced by ROSEN is

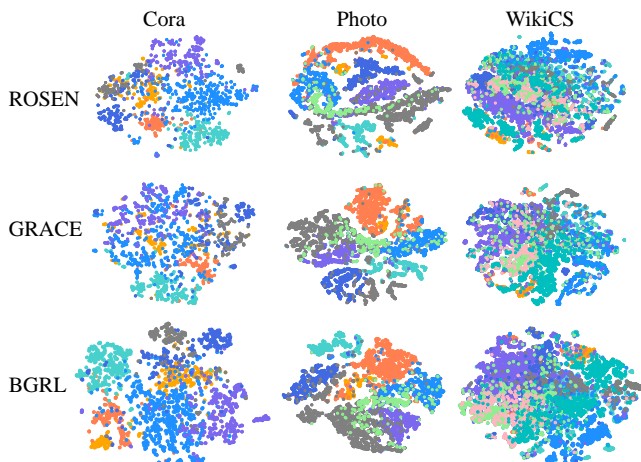

Figure 3: t-SNE visualization of the models GRACE, BGRL, and ROSEN on the datasets Cora, Photo, and Wiki-CS. Each color stands for a class.

significantly better than that of the comparison methods. In particular, across all three datasets, the former has a tighter cluster of same-class embeddings and larger gaps between different-class clusters compared to the latters. These results illustrate the discriminative quality of embedding generated by the proposed ROSEN and validate the powerful data mining capabilities of the ROSEN.

**Node Clustering.** For assessing the ability of GCL methods to generate discriminative node representations, their impact on the clustering task was analyzed on the Cora and Citeseer datasets. In line with the training-then-clustering paradigm, after the model training converges, the learned node representations are evaluated for the representation quality with the help of K-Means. Table 3 exhibits the comparison result of the proposed ROSEN and four classical unsupervised baseline methods (i.e., K-Means [10], GRACE [44], BGRL [29], and MVGRL [11]) on the node clustering task.

As can be observed from the data presented in Table 3: (1) compared to the unsupervised method K-Means, which does not use graph structure, considering the homophilous graph structure in the model design enables GCL methods to generate more discriminative representations. (2) compared to these representative GCL methods, the proposed ROSEN has learned clustering-friendly embeddings. It outperforms the runner-up MVGRL by over at least 1% on all datasets, which highlights the effectiveness of leveraging reliable graph structure in the optimization of GCLs.

## 4.3 Effectiveness Test

As discussed in the previous section, the pursuit of graph structure matrices that meet the block diagonal property is proposed to efficiently construct homophilous graph structures in self-supervised scenarios. To visually explain the usefulness of the proposed graph structure inference module, the edge distributions of the inferred graph structure matrices (i.e., $A_Z$ or $A_B$) and the adjacency matrix are compared, as depicted in Figure 4. To examine for the block diagonal property, all nodes are numbered according to their class. And then, the sum of edges connected to nodes of different classes

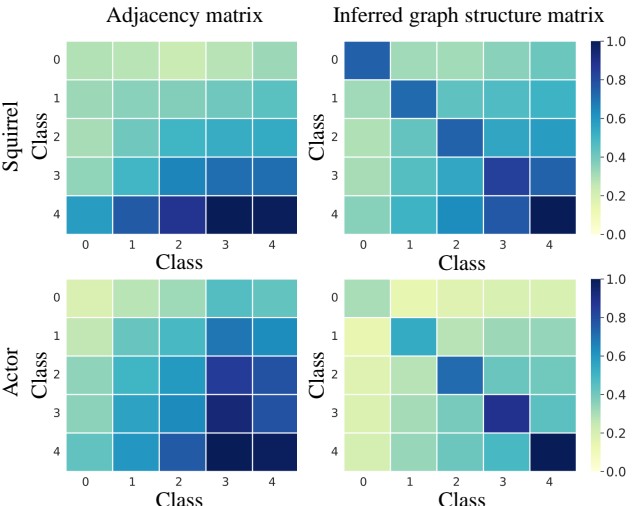

Figure 4: Visualization of the adjacency matrix and the inferred graph structure matrices on the Squirrel and Actor datasets. The inferred graph structure matrices are more similar to the block diagonal matrix.

Table 4: The accuracy in percentage (mean ± std) of node classification for ablation study. The best results are in bold, and the second-best results are underlined.

|  | Cora | Cornell | Wisconsin |
|---|---|---|---|
| ROSEN | $87.72_{\pm1.00}$ | $76.49_{\pm6.84}$ | $78.63_{\pm4.68}$ |
| w/o being in encoding | $85.77_{\pm1.49}$ | $69.08_{\pm6.48}$ | $74.81_{\pm4.96}$ |
| w/o being in contrasting | $85.63_{\pm1.57}$ | $67.24_{\pm5.67}$ | $72.98_{\pm6.19}$ |
| localized GCL | $84.65_{\pm1.18}$ | $56.62_{\pm6.85}$ | $63.02_{\pm5.81}$ |

is counted. In the end, maximum value normalization makes the data map from 0 to 1.

Observing Figure 1 we can notice: the inferred graph structure matrices are always closer to the block diagonal matrices than the original adjacency matrices. In particular, as exemplified on the Squirrel dataset, many elements of the non-diagonal are deleted or the weights are reduced, which illustrates the outstanding ability of the proposed graph structure inference module. Besides that, this phenomenon is more obvious on the Actor dataset, which illustrates the validity of the proposed ROSEN.

## 4.4 Ablation Study

To verify the contributions of the design in the proposed ROSEN, several ablation experiments are constructed on the Cora, Cornell, and Wisconsin datasets. The influence of the manner in which the graph structures are employed, i.e., encoding and contrasting, is studied on the node classification performance of the proposed ROSEN. The results are exhibited in Table 4, where the "localized GCL" stands for the GCL variants in which the adjacency matrix is utilized for both encoding and contrasting.

It can be observed from Table 4 : (1) Compared to other variants, the proposed ROSEN consistently presents optimal classification performance on all datasets, which underlines the effectiveness of its design. (2) The second row presents higher results than those

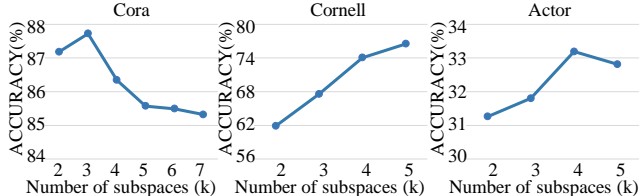

**Figure 5: Impact of the number of diagonal blocks (i.e., number of subspaces) on the performance of node classification.**

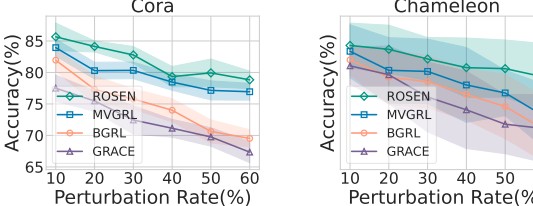

**Figure 6: Performance variation of GCL models on graph data with topology noise.**

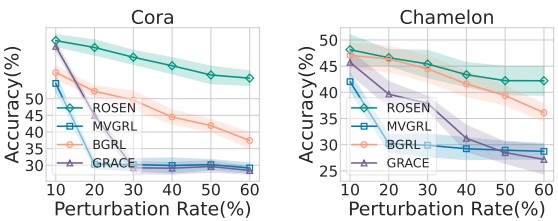

**Figure 7: Performance variation of GCL models on graph data with attribute noise.**

presented in the fourth row. This illustrates the validity of the starting point of this study, i.e., to provide a reliable graph structure for contrastive loss to determine positive and negative samples. (3) The results in the first row are higher than those in the second row, which is attributed to the efficiency of the graph structure in the encoding for generating low-noise representations as a tool for helping the correct blocking in the proposed graph structure inference module. (4) In the comparison of the results in the third and fourth rows, the former over the latter exhibits performance advantages. It highlights the fact that generating discriminative node representations by leveraging the structures inference module is an effective solution for improving GCL.

### 4.5 Hyperparameter Analysis

A unique and crucial parameter in the ROSEN is the number of subspaces (blocks), i.e., $k$, which is ideally the same as the number of classes. Thus, selecting the appropriate number of subspaces can be challenging for self-supervised tasks. This experiment attempts to provide valuable insights by analyzing the effect of this parameter on the node classification performance of the proposed ROSEN, as described in Figure 5. Since there is massive noise in web graphs, the choice of this parameter is limited to no more than the number of classes to preserve sufficient neighbor nodes for each node.

As can be observed from Figure 5: (1) On all three datasets, the proposed ROSEN exhibits steady improvements achieved within a certain range of the parameter $k$, which are $\{2, 3\}$, $\{2, 3, 4, 5\}$ and $\{4, 5\}$ for the Cora, Cornell, and Actor datasets, respectively. This illustrates the insensitivity of the proposed ROSEN to the number of blocks $k$. (2) Together with the data in Tables 1 and Tables 2, it is apparent that the proposed ROSEN still outperforms most baseline methods with small parameter values, such as $k = 2$. The reasonable explanation for this phenomenon is that compared to existing GCL methods which treat all neighbors as positive samples, even if FALSE positive samples (i.e., nodes of different classes) are incorrectly preserved during structure inference, the proposed ROSEN benefits from its exclusion of the nodes of part of the other classes. The results demonstrate the effectiveness of the proposed graph structure inference module for GCL.

### 4.6 Robustness Analysis

As stated in Section 3.1.1, the graph structure inferred from the local neighborhoods endows the adaptability of the ROSEN to noisy data. To verify the claim, this experiment simulates real noisy data by manually setting topology noise and attribute noise for datasets Cora and Actor, as a way of comparing the robustness of the ROSEN and the compared methods (including GRACE, MVGRL, adn BGRL).

It can be observed from Figure 6 that, while the compared methods manifest the adaptability to minor topology noise, their performance degrades incrementally as the perturbation increases. By contrast, the proposed ROSEN exhibits the stability in preserving its predictive performance on graphs with topology noise. In addition, it can be seen from Figure 6 that the proposed model shows the greatest robustness, despite a decline in performance for all models as the proportion of attribute noise increases. Of interest here is on both datasets, the performance reduction of the ROSEN is less than 8%, even with a 60% proportion of noise being added. Similarly, the results in Figure 7 show that the proposed method has better stability than the compared methods. This is mainly attributed to the usefulness of the block constraints imposed on the graph structure matrix. It forcibly tightens the representation of nodes with high feature similarity into the same block, thereby the proposed ROSEN is insensitive to the number of edges.

## 5 CONCLUSION

In summary, this study discusses challenges in self-supervised learning for diverse web graph data and proposes a solution called ROSEN. Existing GCL frameworks face issues related to blind positive sample collection and pseudo-supervised information decay. ROSEN aims to address these problems using a Block Diagonal graph structure inference module and two strategies: "Local Feature Space Training" and "Alternating Update". Specifically, "Local Feature Space Training" refers to a training approach that focuses on the features or characteristics of data within a localized or specific region. Moreover, it benefits from reciprocal information by alternately updating the inferred block diagonal graph structure, and contrastive loss optimization. Overall, this research offers a promising solution to enhance the universality of GCL to diverse web graphs. The potential future research directions include the model design for large-scale web graph applications.

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

# A APPENDIX

## A.1 Datasets

The statistics of all the datasets used in this experiment are shown in Table 5.

**Table 5: Statistics of twelve graph datasets. The abbreviation #EdgeHom denotes the edge homophily elucidated in [26].**

| Dataset | Nodes | Edges | Features | Classes | #EdgeHom |
|---|---|---|---|---|---|
| Cora | 2,708 | 5,278 | 1,433 | 7 | 0.81 |
| CiteSeer | 3,327 | 4,552 | 3,703 | 6 | 0.74 |
| PubMed | 19,717 | 44,324 | 500 | 3 | 0.80 |
| Wiki-CS | 11,701 | 216,123 | 300 | 10 | 0.65 |
| Computers | 13,752 | 245,861 | 767 | 10 | 0.78 |
| Photo | 7,650 | 238,163 | 745 | 8 | 0.83 |
| Cornell | 183 | 295 | 1,703 | 5 | 0.13 |
| Texas | 183 | 309 | 1,703 | 5 | 0.11 |
| Wisconsin | 251 | 499 | 1,703 | 5 | 0.20 |
| Chameleon | 2,277 | 36,101 | 2,325 | 5 | 0.23 |
| Squirrel | 5,201 | 217,073 | 2,089 | 5 | 0.22 |
| Actor | 7,600 | 33,544 | 931 | 5 | 0.22 |

## A.2 Algorithm description

To jointly modify the graph structure $S$ and train the GCL parameters $\Theta$, we alternately update one while fixing the other, and the details are shown in Algorithm 1. And we show in Alogrithm 2 that the structure of graph is inferred through the node embedding.

---

**Algorithm 1** ROSEN

**Input:** Graph $G = (V, E, X)$, GNN encoder $f_\Theta$, number of subspaces $k$
**Output:** trained GNN encoder $f_\Theta$ and node embeddings $H^*$
1: Initialize GNN encoder $f_\Theta$, and initialize $H^* = X$ and initialize augmented graph $S^*$ via Equation (7)
2: **while** not Converge **do**
3:    /* E-step */
4:    $\hat{G} = (\hat{V}, \hat{E}, \hat{X}) = Aug(G(V, E, X))$
5:    **while** not MaxEpoch **do**
6:      $H^* = f_\Theta(S^*, H^*)$
7:      **for** $v$ in $V$ **do**
8:        $H_v^* = H^*[N(v)]$
9:        $Z, B$.append(StructureInference($H_v^*, k$))
10:      **end for**
11:     Update $S^*$ via Equation (13)
12:    **end while**
13:    /* M-step */
14:    **while** not MaxEpoch **do**
15:      $\mathcal{L}_{contrast}(H^*, S^*)$
16:      $\theta = Adam(\mathcal{L}_{contrast}, \Theta)$
17:    **end while**
18: **end while**

---

## A.3 Theoretical Analysis

THEOREM A.1. *Let $\Theta$, $k$ and $\mathbb{1}G$ denote the parameters of the GNN encoder, the number of subspaces (blocks), and the subspace indicator, respectively, the proposed ROSEN follows the Expectation-Maximization (EM) algorithm, in which the structure inference and the maximization of the lower bound on the mutual information of the contrastive pairs' representation are equivalent to the E step and the M step, respectively.*

PROOF. In the localized GCL framework with the variable positive sample set, the optimal parameters of the GNN encoder is learned by maximizating the function $\mathcal{L}(\Theta, \Omega)$. This can be formulated as

$$\Theta^* = \arg\max_\Theta \sum_{v \in V} \log \sum_{u \in N(v,\Omega)} p(h_v, h_u | \Theta) \quad (15)$$

However, since the latent variables, the directly computation of Equation 15 is difficult. With the help of Amortized Variational Inference [34], this problem can be alleviated by introducing the approximated posterior $p(h_u | h_v, \Theta)$. Therefore, the log-likelihood function $\mathcal{L}(\Theta, \Omega)$ in Equation 15 can be reformulated as

$$\mathcal{L}(\Theta, \Omega) = \sum_{v \in V} \log \sum_{u \in N(v,\Omega)} p(h_u | h_v, \Theta) \frac{p(h_v, h_u | \Theta)}{p(h_u | h_v, \Theta)}$$
$$\geq \sum_{v \in V} \sum_{u \in N(v,\Omega)} p(h_u | h_v, \Theta) \log p(h_v, h_u | \Theta) \quad (16)$$
$$- p(h_u | h_v, \Theta) \log p(h_u | h_v, \Theta)$$

where the inequality holds due to the Jensen's inequality. It is worth noting that $-p(h_u | h_v, \Theta) log p(h_u | h_v, \Theta)$ is the entropy operator, which does not affect the update of the parameter $\Theta$. Overall, the log-likelihood function can be formulated as

$$l = \sum_{v \in V} \sum_{u \in N(v,\Omega)} \log p(h_u | h_v, \Theta) p(h_v, h_u | \Theta) \quad (17)$$

Accordingly, the EM algorithm for optimizing this function can be described as inferencing structure in E step and maximizating the lower bound on the mutual information in M step.
**E step.** To infer the approximated posterior probability $p(h_u | h_v, \Theta)$, the proposed structure inference module introduce the bloack diagonal constraint. Thus, this probability can be formulated as

$$p(h_v | h_u, \Theta) = \sum_{t=1}^{k} p(h_u | h_v, t, \Theta) p(t | h_v, \Theta) \quad$$

It can be obtained from the local block diagonal graph structure inference in Section 3.1.1, namely $p(h_v | h_u, \Theta) = \mathbb{1}G_{v,u}$, which assumes that the neighbour nodes which belong to the same subspace are of the same class ($\mathbb{1}G_{v,u} = 1$), i.e., TRUE positive samples.
**M step.** Based on the E step, the M step focuses on maximizating the lower-bound of Equation 17. In particular, there are

$$l = \sum_{v \in V} \sum_{u \in N(v,\Omega)} p(h_u | h_v, \Theta) \log p(h_v, h_u | \Theta) \quad (18)$$
$$= \sum_{v \in V} \sum_{u \in N(v,\Omega)} \mathbb{1}G_{v,u} \log p(h_v, h_u | \Theta) \quad (19)$$

---

**Algorithm 2** Structure Inference

**Input:** Ego embeddings of node $\mathbf{H_i}^*$, number of subspaces $k$
**Output:** The coefficient matrices $\mathbf{Z}$, $\mathbf{B}$
1: Initialize $\mathbf{Z} = \mathbf{B} = \mathbf{W} = 0, \epsilon = 1e - 8$
2: **while** not MaxEpoch **do**
3:      Update $\mathbf{Z_{k+1}}$ by $\mathbf{Z}^{k+1} = \underset{\mathbf{Z}}{\operatorname{argmin}} \frac{1}{2}\|\mathbf{X} - \mathbf{XZ}\|^2 + \frac{\lambda}{2}\|\mathbf{Z} - \mathbf{B}\|^2$
4:      Update $\mathbf{B_{k+1}}$ by $\mathbf{B}^{k+1} = \underset{\mathbf{B}}{\operatorname{argmin}} \frac{\lambda}{2}\|\mathbf{Z} - \mathbf{B}\|^2 + \gamma\langle\operatorname{Diag}(\mathbf{B1}) - \mathbf{B}, \mathbf{W}\rangle$, s.t. $\operatorname{diag}(\mathbf{B}) = 0, \mathbf{B} \geq 0, \mathbf{B} = \mathbf{B}^\top$
5:      Update $\mathbf{W_{k+1}}$ by $\mathbf{W}^{k+1} = \underset{\mathbf{W}}{\operatorname{argmin}}\langle\operatorname{Diag}(\mathbf{B1}) - \mathbf{B}, \mathbf{W}\rangle$, s.t. $0 \leq \mathbf{W} \leq \mathbf{I}, \operatorname{Tr}(\mathbf{W}) = k$
6:      Check the convergence conditions

        $\|\mathbf{Z_{k+1}} - \mathbf{Z_k}\|_\infty \leq \epsilon, \|\mathbf{B_{k+1}} - \mathbf{B_k}\|_\infty \leq \epsilon$
7: **end while**
8: **return** The coefficient matrices $\mathbf{Z}$, $\mathbf{B}$

---

and $p(\mathbf{h}_v, \mathbf{h}_u | \Theta) = \frac{1}{|N(v,\Omega)|} p(\mathbf{h}_u | \mathbf{h}_u, \Theta)$. Since we consider that the prior obeys a uniform distribution, and describe the distribution of each sample in the feature space with isotropic Gaussian, thus there is

$$p(\mathbf{h}_v | \mathbf{h}_u, \Theta) = \frac{1}{2\sigma_i^2} \exp\left(-\frac{1}{2\sigma_1^2}(\mathbf{h}_v - \mathbf{h}_u)^T \cdot (\mathbf{h}_v - \mathbf{h}_u)\right) \quad (20)$$

$$= \frac{1}{2\sigma_i^2} \exp\left(-\frac{\left(\mathbf{h}_v^T \cdot \mathbf{h}_u - 1\right)}{2\sigma_1^2}\right) \quad (21)$$

where the last equivalence due to the L2 normalization for the features $\mathbf{H}$. Setting $\tau = \sigma^2$ as the hyperparameter for all terms, ignoring the constant term, and taking Equation 21 into Equation 19, it can be obtained as

$$\sum_{v \in V} log \frac{\sum\limits_{v^+ \in N_v^S} \mathbb{1}\mathbf{G}_{v,v_+} * e^{\theta(\mathbf{h}_v, \mathbf{h}_{v^+})/\tau}}{\sum\limits_{v^+ \in N_v^S} \mathbb{1}\mathbf{G}_{v,v_+} * e^{\theta(\mathbf{h}_v, \mathbf{h}_{v^+})/\tau} + \sum\limits_{v^- \in \{V \setminus \mathcal{N}_v^S\}} e^{\theta(\mathbf{h}_v, \mathbf{h}_{v^-})/\tau}}$$

It can be discovered that when considering weights which reflect local higher-order relationships, the objective is equivalent to the ROSEN. □

Received 20 February 2007; revised 12 March 2009; accepted 5 June 2009

