# OpenReview forum: "Graph Contrastive Learning Reimagined: Exploring Universality"
_ACM.org/TheWebConf/2024/Conference — TheWebConf24_

### Official Review · Reviewer_f6E3 · 2023-11-22

**Novelty:** 3
**Technical Quality:** 2

**Review:**

This paper introduces a new graph contrastive learning method, named ROSEN, for heterophilous graphs. The authors argue the encoder design and positive sample collection of existing graph contrastive learning are not well-suited for heterophilous graphs. ROSEN overcomes these limitations by investigating the block diagonal properties of homophilous graph structures and employing a self-supervised structure inference module. This module is designed to adapt GCL to generate discriminative node representations through Local Feature Space Training and Alternating Optimization. The framework is theoretically grounded in the expectation-maximization algorithm and has been tested extensively on publicly available datasets.

Pros:
1. Graph contrastive learning is a trending direction with important applications.
2. The use of the expectation-maximization algorithm provides a theoretical foundation for the ROSEN framework, which is interesting.

Cons:
1. The presentation of the paper is poor so I may not understand the proposed method correctly. The authors are strongly advised to revise their manuscript.
2. There exist many works for self-supervised learning for heterophily graphs, such as [1-4], which are not discussed and compared in the paper.
3. In the experiments, the authors only use GCN as backbones and do not compare with methods using heterophily GNNs as backbones.
4. I also wonder what is the complexity or the efficiency of the proposed method, which is not discussed in the paper.
5. The term universality is very confusing. The authors may simply state their proposed method is suitable for both homophily and heterophily graphs.

[1] Towards Self-supervised Learning on Graphs with Heterophily, CIKM’22
[2] Self-supervised Learning and Graph Classification under Heterophily, CIKM’23
[3] Can Single-Pass Contrastive Learning Work for Both Homophilic and Heterophilic Graph, arXiv’22
[4] Contrastive Learning under Heterophily, arXiv’23

**Questions:**

See above

**Reviewer Confidence:**

2: The reviewer is willing to defend the evaluation, but it is likely that the reviewer did not understand parts of the paper

**Scope:**

3: The work is somewhat relevant to the Web and to the track, and is of narrow interest to a sub-community

---

### Official Review · Reviewer_dnBm · 2023-11-23

**Novelty:** 6
**Technical Quality:** 6

**Review:**

> Summarization

This paper investigates the universality of graph contrastive learning. The main difficulties are the homophilous encoder and homophilous positive sampling. Firstly, it presents the infeasibility of the direct employment of a universal encoder. Then, it points out that the main reason can be attributed to the positive sampling, which can’t distinguish inter- and intra-class nodes. To alleviate this issue, this paper observes the block-diagonal property of the graph structure and uses self-expressive subspace clustering to select true positive samples and update the graph topology in the encoder effectively. Finally, it provides a theoretical understanding of the proposed model via its optimization process. Experimental evaluations demonstrate its superior performance on heterophilious graphs.

> Strengths

- The motivation of universal graph contrastive learning by reliable positive sampling makes sense.
- The technical novelty is significant. It is interesting to employ local subspace clustering and block-diagonal property for true positive sampling. The theoretical insight is rigorous.
- The experiment evaluations are convincing. The performance improvements on heterophilious graphs are remarkable. The ablation study and robustness to noises are sufficient.

> Weaknesses

- The presentation should be carefully polished. There are many typos.
- Some experiment analyses are not enough. For example, it is not clear why the proposed model can be robust to noise.
- The theoretical analysis lacks intuitive interpretations. Although the authors provide rigorous proof, the connection between the theorem and the superior performance should be given.

**Questions:**

See weaknesses.

**Ethics Review Description:**

No ethics issue

**Reviewer Confidence:**

3: The reviewer is confident but not certain that the evaluation is correct

**Scope:**

3: The work is somewhat relevant to the Web and to the track, and is of narrow interest to a sub-community

---

### Official Review · Reviewer_ryYt · 2023-11-24

**Novelty:** 6
**Technical Quality:** 5

**Review:**

This paper focuses on self-supervised representation learning on graphs (specifically, nodes). Its motivation is to address the potential issue of false positives in existing local contrastive learning methods. The core idea of this paper is to learn a completely new graph structure that has a higher homophily ratio, which can alleviate the false positive problem to some extent. The authors propose an alternative learning framework that involves optimizing the graph structure based on embeddings and then performing contrastive learning based on the optimized graph structure. Extensive experiments demonstrate the effectiveness of the proposed method in this paper.

Overall, this paper is well-written, the motivation is reasonable, and the proposed method is solid. However, I have some specific implementation-related questions, and I'm not entirely satisfied with the experimental results. Given the current quality of the paper, I find it difficult to provide an acceptance recommendation. I have listed my questions in the 'Questions' section below. I hope the authors can respond to my concerns seriously, and I am willing to increase my score if my concerns are effectively addressed.

**Questions:**

>Q1: What's the implication of the first term in Eq.6? Does it assume feature homophily?

>Q2: Line 419 to Line 427: The complete graph structure is generated as follows: based on the author's description, the proposed method first optimizes the graph structure for each node's ego-net, and then merges the graph structures of these ego-nets. However, considering that if two nodes belong to many different ego-nets, the question of whether these two nodes will generate edges in different ego-nets may have different answers. It is not clear how this situation is resolved.

>Q3: Line 436: What exactly does 'Robust topology augmentation' refer to? I find it difficult to understand the specific implications of this paragraph.

>Q4: I wonder if it's really reasonable to learn the graph structure on ego-nets using the method proposed in the paper. The method described in the paper results in ego-nets producing k disjoint subgraphs, each representing a class. However, the nodes in ego-nets may not necessarily contain nodes from all classes. Considering an extreme case where all nodes in the target node's ego-net belong to the same class, is this way of generating subgraphs harmful?

>Q5: Citation networks: What is the actual partitioning of the citation networks? In lines 514 to 521, the authors state that experiments were conducted using publicly available partitions for Cora, Citeseer, and Pubmed. However, it is widely known that these three datasets have a public split where each class has 20 nodes considered as the training set. Based on the description in the paper, it appears that the authors used a partition of 48%, 32%, and 20%, which is not a common division in the field of self-supervised graph learning. Nevertheless, the results presented in Table 1 are challenging to believe as they seem too low. I suspect that the actual partition might be closer to 1:1:8. Have the authors carefully verified the correct dataset partition, and were the results in Table 1 obtained through independent replication? In my experience, GAE can achieve much better results than those reported in Table 1. Additionally, I suggest that the authors try the public split's results, considering that most baselines are based on the public split.  Furthermore, some crucial baselines have been overlooked [1,2].

References:

[1] Lee, Namkyeong, Junseok Lee, and Chanyoung Park. "Augmentation-free self-supervised learning on graphs." Proceedings of the AAAI Conference on Artificial Intelligence. Vol. 36. No. 7. 2022.

[2] Zhang, Hengrui, et al. "Localized Contrastive Learning on Graphs." arXiv preprint arXiv:2212.04604 (2022).

>Q6: What is the change in the number of edges in the graph obtained through ROSEN? If the number of edges significantly decreases, I believe that even if the learned graph has a high degree of homophily, it may not have much practical significance.

>Q7: Visualization in Figure 3: The authors claim that the quality of node embeddings obtained by ROSEN is significantly better, but I do not see this clearly from Figure 3. At least on Cora, I believe that BGRL's embedding quality is not worse than ROSEN, and on Photo, GRACE's embedding quality is also high. Furthermore, the strip-shaped embedding clusters obtained by ROSEN on Photo do not seem to be a good sign, considering that embeddings are generally used directly for linear classification.

>Q8: Does ROSEN involve pair contrastive learning through data augmentation? If not, I would not recommend the authors to place ROSEN in the context of graph contrastive learning (although I can accept it if they do). In my view, doing contrastive learning in the neighborhood is more akin to graph autoencoders. I just believe that there is no need to introduce pair contrastive learning in Section 2.1 as it is unrelated to this paper.

>Q9: Can the authors provide a training time cost comparison with other methods since the proposed method looks very inefficient?

**Reviewer Confidence:**

4: The reviewer is certain that the evaluation is correct and very familiar with the relevant literature

**Scope:**

3: The work is somewhat relevant to the Web and to the track, and is of narrow interest to a sub-community

---

### Official Review · Reviewer_ojHD · 2023-11-24

**Novelty:** 3
**Technical Quality:** 3

**Review:**

## Summary: ##
The paper addresses the challenge of label scarcity in real-world graph data, particularly focusing on the limitations of Graph Contrastive Learning (GCL) in heterophilous networks. The paper proposes a framework named ROSEN, which leverages block diagonal properties of homophilous graph structures for contrastive learning. This framework theoretically follows the Expectation Maximization (EM) algorithm and has been tested on publicly available datasets to demonstrate its effectiveness and universality.

## Strength: ##
1. The research topic is interesting and important for the graph machine learning community.
2. The paper is relatively well-written and easy to read, although relatively complex techniques are proposed.
3. The experiments show the effectiveness of the method.

## Weaknesses: ##
- The experiments should be conducted on more realistic benchmarks such as the Open Graph Benchmark.
- More intuitive analyses should be present for the experimental results as well as the figures to support the main points in the paper.
- No selected hyperparameters of the method are reported in the paper.

**Questions:**

Besides weaknesses above, I am curious about the efficiency of the method.

**Ethics Review Description:**

There are no ethics concerns.

**Reviewer Confidence:**

4: The reviewer is certain that the evaluation is correct and very familiar with the relevant literature

**Scope:**

3: The work is somewhat relevant to the Web and to the track, and is of narrow interest to a sub-community

---

### Decision · Program_Chairs · 2024-01-22

**Decision:**

Accept

**Comment:**

This paper is improving upon existing Graph Contrastive Learning (GCL) methods by introducing a novel positive sampling module which relies on the observation that homophilous graphs are associated with block-diagonal adjacency matrix structures, which are thus enforced in inferring a feature-induced graph which is used for positive sampling. The paper provides theoretical and empirical results for the claims made within.

 The reviews by and large recognize the novelty of the proposed method.

 Furthermore, even though most reviews raise some questions regarding the experimental evaluation, those question seem to be adequately and favorably answered during the rebuttal process.

 There are a number of less critical points raised by some reviews, especially pertaining to improving the intuitive connections between theoretical results and observed performance and improving the overall presentation of the work.

 The authors should take into account all points of discussion and incorporate them into the final version of the paper.